# Assessing the impact of color blindness on the ability of identifying benign and malignant skin lesions by naked-eye examination

**Mutasem Elfalah**[1], **Nesrin Sulyman**[2], **Anas Alrwashdeh**[3], **Sari Al Hajaj**[4], **Sonia Alrawashdeh**[5], **Asad Al-Rawashdeh**[6], **Saif Aldeen AlRyalat**[1]\*

1 Department of Ophthalmology, The University of Jordan, Amman, Jordan, 2 Internal Medicine, Primary Healthcare Corporation, Doha, Qatar, 3 Internal medicine, King Hussein Cancer Center, Amman, Jordan, 4 Medical Doctor, School of Medicine, The University of Jordan, Amman, Jordan, 5 Department of Ophthalmology, Saratoga Vitreo-Retinal Ophthalmology, New York, New York, United States of America, 6 Internal Medicine, Private Clinic, Lubbock, Texas, United States of America

\* s.alryalat@ju.edu.jo, saifryalat@yahoo.com

## Abstract

### Background

Color vision deficiency describes the inability to distinguish certain shades of color. The aim of this study was to assess the impact of having color vision deficiency on the accuracy of distinguishing benign and malignant skin lesions by naked-eye examination.

### Methods

This was a cross-sectional study conducted during the period August 2020 to February 2021. We randomly selected a total of 20 nevi and 20 melanoma images from an open access image database. The 40 images were divided into four sets of images, each set contained 5 benign and 5 malignant skin lesion images simulated as if they were seen by a protanope physician, deuteranope physician, tritanope physician, and a set of images presented without simulation. In an online survey, students who were in their final year of medical school or had newly graduated were asked to diagnose each image as benign or malignant.

### Results

A total of 140 participants were included with a mean (SD) age of 24.88 (1.51). We found a significantly higher mean accuracy for non-simulated images compared to deuteranope simulated images (p< 0.001, mean difference = 11.07, 95% CI 8.40 to 13.74). We did not find a significant difference in accuracy classification for protanope simulated images (p = 0.066), nor for tritanope simulated images (p = 0.315). Classification accuracy for malignant lesions was higher than classification accuracy for benign lesions, with the highest difference belonging to deuteranope simulated images, with a difference in mean accuracy of classifying malignant lesions by 32.2 (95% CI 27.0 to 37.6).

**Data Availability Statement:** This study used the MED-NOD dataset, an open access skin lesion image dataset available at their website: http://

www.cs.rug.nl/~imaging/databases/melanoma_naevi/. The images used in this study, including simulated images, were deposited in Harvard dataverse at https://doi.org/10.7910/DVN/OX324U.

**Funding:** The author(s) received no specific funding for this work.

**Competing interests:** The authors have declared that no competing interests exist.

## Conclusion

Deuteranope participants (i.e., green color deficiency) had a significantly lower accuracy of distinguishing pigmented skin lesions as benign or malignant, an impact not found for other color vision deficiencies, which was mainly for misdiagnosing benign lesions as malignant.

## Introduction

It has been almost 35 years since the introduction of the ABCD pneumonic to simplify and facilitate the detection and early recognition of malignant pigmented skin lesions [1]. The pneumonic originally described suspicious characteristics that may warrant referral to specialized dermatologists, including "A" for asymmetry, "B" for border irregularity, "C" for color variegation, and "D" for a diameter of more than 6 mm. Since then, several expert groups revisited the criteria, trying to increase its accuracy, by avoiding falsely classifying benign lesions as malignant (false positive), and more importantly, avoiding classifying malignant lesions as benign (false negative) [2–4]. They have also added the letter "E", referring to evolution, a rapid change in shape, color, or size of the lesion [2]. Despite the need to put all lesion characteristics in the context, the color criterion was found to be among the top of the ABCDE criteria regarding accuracy of discriminating malignant lesions [4, 5]. Moreover, lesion pattern recognition, which mainly involved the color of the lesion, also played an important role in malignant lesion detection [6, 7].

Color vision deficiency originally described the inability to distinguish certain shades of color. On the other hand, the term "color blindness" is sometime used interchangeably [8]. Despite its relative high prevalence, people might be unaware of their color vision deficiency [9]. Physicians and healthcare professionals might also be affected by color vision deficiency without being aware of their condition, which might have a significant impact on their practice [10]. In a previous study, skin condition assessment was among the most difficult tasks for people with color vision deficiency [11].

Patients with skin lesions usually present to primary care physicians for evaluation, those lesions thought to be of malignant potential are usually referred to dermatology clinics for further evaluation and possible intervention. The diagnostic and referral accuracy of pigmented skin lesions were generally found to be low among primary care physicians compared to dermatologists [12–14]. The impact of having color vision deficiency on distinguishing benign and malignant skin lesions has never been studied, despite the potential impact on lesion color, an integral part of the ABCDE criteria. The aim of this study was to assess the impact of having color vision deficiency on the accuracy of distinguishing benign and malignant skin lesions by naked-eye examination. We conducted this study on participants who were in their final year in medical school or had newly graduated, whom we believe depend more on lesion color criteria in distinguishing benign and malignant lesions.

## Methods and materials

### Design and participants

This was a cross-sectional study conducted during the period between August 2020 and February 2021. The study was approved by the institutional review board (IRB) and research committee at Jordan university hospital and the University of Jordan, and was conducted in concordance of the latest declaration of Helsinki. The study included participants who were

either in their final year of medical school or had newly graduated and were working as general practitioners. Participants who did not have a dermatology rotation during their medical school were excluded at the beginning of the survey.

Using an online form, we detailed the nature of the project. Participants then provided their consent for participation. The rater was first asked to take an online Ishihara test to confirm the absence of preexisting color vision deficiency, and participants who had a score below 12/13 were asked not to proceed with the questionnaire. After that, the rater was provided a short online presentation on Medscape medical platform that explains how to differentiate between benign and malignant tumors [15].

## Assessment

The questionnaire started by asking the participant if he/she took a dermatology rotation during medical school, where all Jordanian universities usually have a 2-week long dermatology rotation with an exam at the end of rotation to assess acquired knowledge. This was followed by Ishihara testing, as described above. After that, the questionnaire asked about the participant's demographic variables (i.e., age, gender, and occupation).

Regarding benign and malignant lesions used in the survey, we used randomizer.org to randomly select a total of 20 nevi and 20 melanoma images from MED-NODE database [16]. The 40 images were divided into four sets of images, each set contain five images for benign and five images for malignant skin lesion. The four sets were:

- Five benign and five malignant skin lesions to be simulated as if were seen by a protanope physician;

- Five benign and five malignant skin lesion images to be simulated as if were seen by a deuteranope physician;

- Five benign and five malignant skin lesion images to be simulated as if were seen by a tritanope physician.

- Five benign and five malignant skin lesion images presented without simulation.

To transform images into what a protanope, deuteranope, and tritanope participant would see, we used Vischeck color blindness simulator implemented in Fiji software [17], which has been proven to be highly accurate in simulating color-blind images [18]. We used Google forms for data collection. In order to correct for any variabilities in the screens of the device's, we compared the accuracy of non-simulated images with simulated images. (Fig 1) shows a benign skin lesion (i.e., nevus) simulated into what a protanope, deuteranope, and tritanope participant would see, and (Fig 2) shows a malignant skin lesion (i.e., superficial spreading melanoma) simulated into what a protanope, deuteranope, and tritanope participant would see. Both figures were not part of the MED-NODE images used in this study's survey. The 40 images used in this study's survey were deposited openly in Harvard Dataverse [19]. It is important to note that both images were collected from our hospital setting after proper consent had been obtained, so they were not a part of the images shown in the survey originally collected for this study. The 40 images were shuffled and were presented sequentially and individually. Images were standardized in size and resolution. The rater was asked to rate each image as either benign or malignant.

## Statistical analysis

We used SPSS 26.0 in our statistical analysis. We used mean and standard deviation (SD) to describe continuous variables, and we used frequency (percentage) to describe categorical

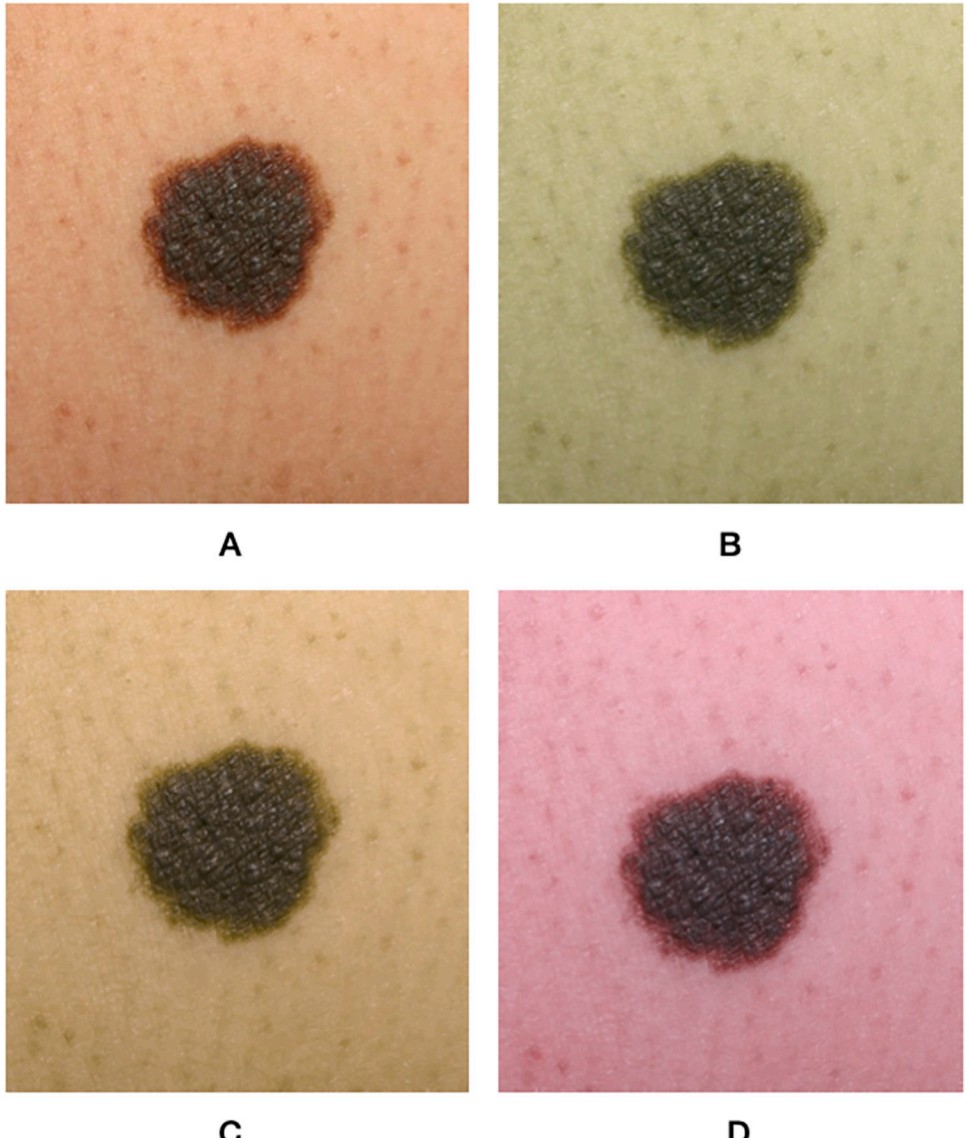

**Fig 1.** A nevus presented as an example of a benign skin lesion (A) simulated into what a protanope (B), deuteranope (C), and tritanope (D) participant would see. Note that the simulation included the lesion and the background skin, affecting the distinction at lesion borders. The image presented here is not a part of the MED-NODE database.

variables. The diagnosis for each image was scored as either correct (1) or incorrect (0). There were four sets of images, each containing images for five benign and five malignant lesions. Of the four sets, 3 were simulated into what a deuteranope, protanope, and tritanope participant would see, and the last set was not simulated. We calculated the accuracy score as a percentage for the benign and malignant images in each set, then the accuracy of image classification in the set (i.e., overall accuracy). We used Paired sample t-test to analyze the mean difference in accuracy between the four sets. We used independent sample t-test to analyze the accuracy score difference between men and women. We presented the data as mean difference and 95% confidence interval (CI). We used a p value threshold of 0.05 as our significance threshold.

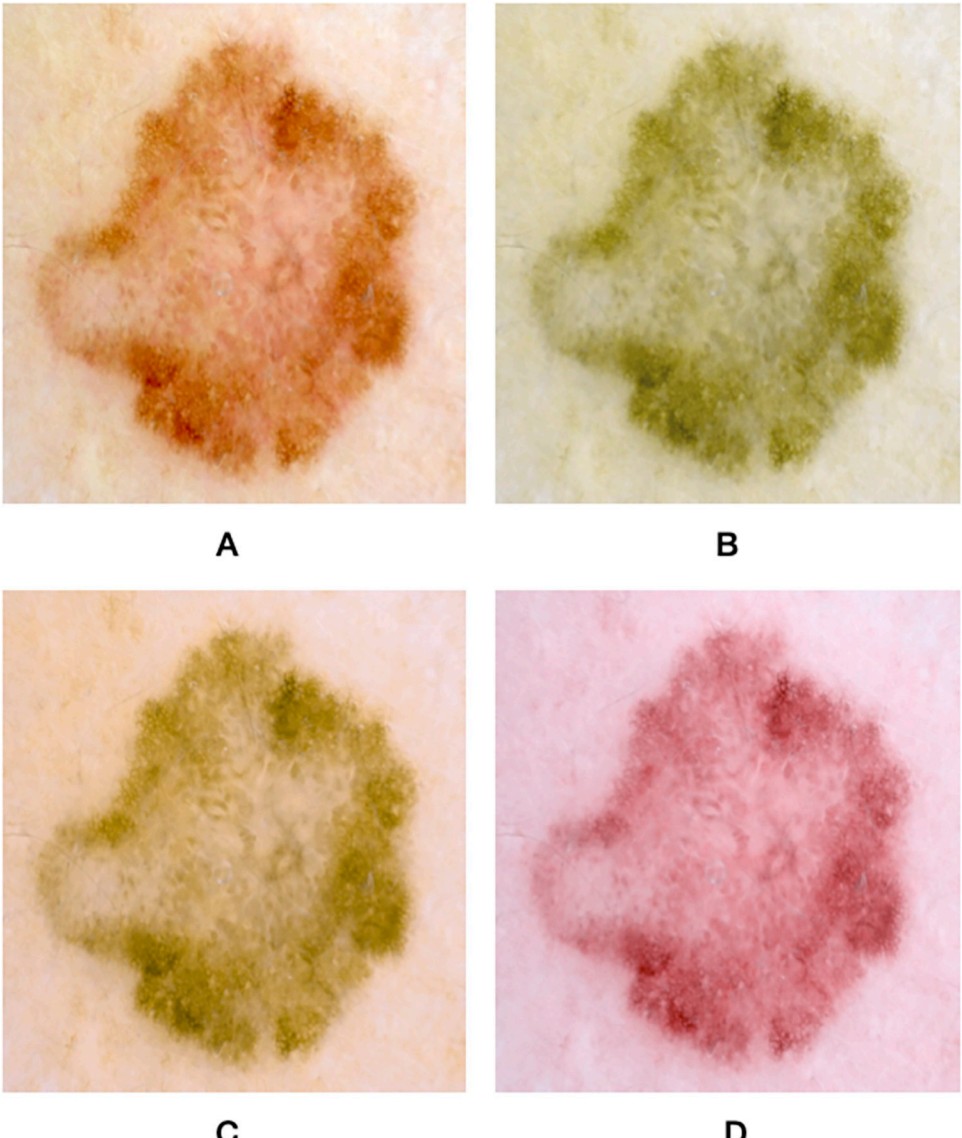

**Fig 2.** A melanoma presented as an example of a malignant skin lesion (A) simulated into what a protanope (B), deuteranope (C), and tritanope (D) participant would see. The image presented here is not a part of the MED-NODE database. Figure adapted from Longo et al. study (non-simulated image) [20].

## Results

A total of 152 participants completed image classification. Among them, 12 participants did not have a dermatology rotation during their medical school, and thus were excluded from further analysis. For the 140 participants included in the analysis, the mean (SD) age was 24.88 (1.51). Among the participants were 86 men with a mean (SD) age of 25.2 (1.50), and 54 women with a mean (SD) age of 24.37 (1.40). Table 1 details the mean accuracy of classifying lesions presented in the images in each set (i.e., protanope, deuteranope, tritanope simulated images and non-simulated images) as either benign or malignant lesions. It also presents the mean difference in classification accuracy between images containing benign and malignant lesions.

**Table 1. The mean accuracy of classifying lesions presented in the images in each set (i.e., protanope, deuteranope, tritanope simulated images and non-simulated images) as either benign or malignant lesions.** As well as, the mean difference in classification accuracy between images containing benign and malignant lesions.

| | | Mean % | Std. Deviation % | p value | Mean difference % | 95% CI % |
|---|---|---|---|---|---|---|
| Protanope simulated images | Accuracy of benign lesion classification | 73.86 | 24.68 | <0.001 | 8.4 | 4.0 to 12.8 |
| | Accuracy of malignant lesion classification | 82.28 | 18.90 | | | |
| Deuteranope simulated images | Accuracy of benign lesion classification | 48.28 | 22.5 | <0.001 | 32.2 | 27.0 to 37.6 |
| | Accuracy of malignant lesion classification | 82.58 | 19.04 | | | |
| Tritanope simulated images | Accuracy of benign lesion classification | 70.0 | 26.98 | <0.001 | 8.4 | 3.0 to 14.0 |
| | Accuracy of malignant lesion classification | 78.42 | 21.6 | | | |
| Non-simulated images | Accuracy of benign lesion classification | 67.14 | 26.06 | <0.001 | 16.7 | 10.6 to 22.8 |
| | Accuracy of malignant lesion classification | 83.86 | 22.5 | | | |

Upon comparing classification accuracy among protanope, deuteranope, and tritanope simulated images with non-simulated images, we found a significant difference in accuracy classification between deuteranope simulated images compared to non-simulated images ($p < 0.001$), with higher mean accuracy for non-simulated images (mean difference = 11.07, 95% CI 8.40 to 13.74). We did not find a significant difference in accuracy classification between protanope simulated images with non-simulated images (p = 0.066), nor between tritanope simulated images with non-simulated images (p = 0.315).

Upon comparing classification accuracy for benign and malignant lesions between each image simulation, we found significant differences for all images, where classification accuracy for malignant lesions was higher than classification accuracy for benign lesions. The highest difference was for deuteranope simulated images, where the accuracy of classifying malignant lesions was higher by a mean of 32.2 (95% CI 27.0 to 37.6), as shown in Table 1.

Upon comparing gender difference in regard to classification accuracy, there was no statistically significant difference between men and women for all groups, as shown in Table 2.

## Discussion

This study tested the accuracy of classifying benign and malignant pigmented skin lesions among images simulated to appear as if seen by participants with different types of color vision deficiency, and we compared their accuracy with images that were not simulated. We found that the only color vision deficiency that affected classification accuracy was deuteranopia (i.e., green color deficiency). Upon performing our sub-group analysis to analyze if there was a difference in classification accuracy for benign or malignant lesions, we found that the classification accuracy for benign lesions was significantly lower for all images, with the highest

**Table 2. Comparison in the accuracy of classifying lesions presented in the images in each set (i.e., protanope, deuteranope, tritanope simulated images and non-simulated images) between male and female participants.**

| | Gender | N | Mean % | Std. Deviation % | p value |
|---|---|---|---|---|---|
| Protanope simulated images | Male | 86 | 79.42 | 18.68 | 0.257 |
| | Female | 54 | 75.93 | 15.96 | |
| Deuteranope simulated images | Male | 86 | 63.49 | 14.61 | 0.301 |
| | Female | 54 | 65.93 | 11.58 | |
| Tritanope simulated images | Male | 86 | 74.65 | 19.08 | 0.720 |
| | Female | 54 | 73.52 | 16.62 | |
| Non-simulated images | Male | 86 | 73.49 | 16.36 | 0.058 |
| | Female | 54 | 78.70 | 14.54 | |

magnitude arising from deuteranopia simulated images. In fact, non-simulated images had lower accuracy for the classification of benign lesions compared to protanopia and tritanopia simulated images. It was noted that the borders between the lesion and the surrounding skin is less distinct for deuteranopia-simulated images, which might increase the rater's classification bias toward malignant lesions. This effect was also observed, but with a lesser degree, for protanopia-simulated images, but not for tritanopia-simulated images. This can be explained by previous studies' observations, where the distinction of image borders increases for higher wavelength colors (i.e., toward the red colors) [21, 22], and deuteranopia- and protanopia-simulated images have more low-wavelength colors than normal or tritanopia-simulated images.

Due to the shortage of specialized dermatology clinics to diagnose and manage skin lesions, primary care physicians may need to depend on their skills and knowledge to diagnose and possibly conduct a biopsy on skin lesions [23]. Ensuring that those physicians can distinguish such lesions is of great importance. Here, we found that physicians with color vision deficiency will have a significantly higher frequency of false positives in their diagnosis', thus, referring and performing procedures more than people without color vision deficiency, particularly those with green color deficiency. It is important to note that green color deficiency is the most common form of color vision deficiency [8].

There are various medical specialties to which a patient with a skin lesion can be presented to, including dermatology and plastic surgery, but most patients end up presenting to primary care clinics [24]. Primary care physicians play an important role in distinguishing skin lesions that warrant referral to specialized clinics, or even lesions that might need clinic based interventions [25, 26]. Many efforts have been evaluated for improving the diagnostic accuracy of primary care physicians for diagnosing malignant melanoma such as short training courses [13, 27, 28]. The most recent reviews of literature indicate that primary care physicians had sensitivities, specificities, and diagnostic accuracies ranging from 0.25 to 0.88, 0.26 to 0.71, and 0.49 to 0.80, respectively in comparison to 0.74 to 1.00, 0.56 to 0.95, and 0.85 to 0.89 sensitivities, specificities, and diagnostic accuracies for dermatologists [12]. Early referral to specialized dermatology clinics will lead to a decrease in the time from first diagnosis to excision and final histopathological diagnosis [29, 30], which in turn has significant impact on survival [31]. While the accuracy of distinguishing benign and malignant lesions was lower with color vision deficiency, it resulted in more lesions categorized as malignant and required further evaluation, rather than missing malignant lesions.

Even though the study included a relatively high number of participants with 40 instances from the openly accessible Med-NODE database to be assessed by each participant [32], with color vision deficiency simulation using high fidelity validated software, it still has limitations that need to be considered upon result's interpretation. While we provided a short course for each participant on how to distinguish benign and malignant features, clinical decision on whether a lesion is benign or malignant is usually also dependent on clinical data, which was not provided in our case. Moreover, this study was not performed on specialized dermatologists, whom would have more experience and confidence in diagnosing benign lesions as benign depending on more lesion characteristics, other than color. Further studies in this regard might show if including experienced dermatologist might abolish such accuracy difference for participants with color vision deficiency. Another limitation to keep in mind is that participants used their own devices to complete the questionnaire, which might result in an impact from screen color and brightness on the classification accuracy.

## Conclusion

Deuteranopia (i.e., green color deficiency) led to a significant decrease in accuracy of distinguishing pigmented skin lesions as benign or malignant on simulated images, an impact that

was not found for other color vision deficiencies. We also found that the poor distinguishing accuracy was mainly for misdiagnosing benign lesions as malignant, which was also most prominent for deuteranopia simulated images.

## Author Contributions

**Conceptualization:** Saif Aldeen AlRyalat.

**Data curation:** Sari Al Hajaj, Sonia Alrawashdeh, Saif Aldeen AlRyalat.

**Formal analysis:** Saif Aldeen AlRyalat.

**Methodology:** Mutasem Elfalah, Nesrin Sulyman, Anas Alrwashdeh, Sari Al Hajaj, Sonia Alrawashdeh, Asad Al-Rawashdeh, Saif Aldeen AlRyalat.

**Project administration:** Mutasem Elfalah.

**Supervision:** Mutasem Elfalah, Saif Aldeen AlRyalat.

**Visualization:** Nesrin Sulyman, Anas Alrwashdeh, Sari Al Hajaj, Sonia Alrawashdeh, Asad Al-Rawashdeh, Saif Aldeen AlRyalat.

**Writing – original draft:** Nesrin Sulyman, Anas Alrwashdeh, Sari Al Hajaj, Sonia Alrawashdeh, Asad Al-Rawashdeh.

**Writing – review & editing:** Mutasem Elfalah, Nesrin Sulyman, Anas Alrwashdeh, Sonia Alrawashdeh, Asad Al-Rawashdeh, Saif Aldeen AlRyalat.

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
