## [Decision Letter · Decision Letter 0]

4 Aug 2021

PONE-D-21-20492

Assessing the impact of color blindness on the ability of identifying benign and malignant skin lesions

PLOS ONE

Dear Dr. AlRyalat,

Thank you for submitting your manuscript to PLOS ONE. After careful consideration, we feel that it has merit but does not fully meet PLOS ONE’s publication criteria as it currently stands. Therefore, we invite you to submit a revised version of the manuscript that addresses the points raised during the review process.

While understanding the impact of color blindness on the ability of diagnosing skin lesions, both clinically and dermoscopically, is certainly of major importance for the dermatology clinic, a study like this is only of use if (1) the right methodology is applied and (2) the data are interpreted without any bias. Thus, in accordance with the expert reviewers I have some conceptual and technical concerns that need to be addressed. Rather than repeat further points here, I refer you to the specific remarks (below) for details. 

We look forward to receiving your revised manuscript.

Kind regards,

Nikolas K. Haass, MD/PhD

Academic Editor

PLOS ONE

Journal Requirements:

2. We note that Figure 1 in your submission contain copyrighted images. All PLOS content is published under the Creative Commons Attribution License (CC BY 4.0), which means that the manuscript, images, and Supporting Information files will be freely available online, and any third party is permitted to access, download, copy, distribute, and use these materials in any way, even commercially, with proper attribution. For more information, see our copyright guidelines: http://journals.plos.org/plosone/s/licenses-and-copyright.

a) You may seek permission from the original copyright holder of Figure 1 to publish the content specifically under the CC BY 4.0 license. 

Additional Editor Comments:

While understanding the impact of color blindness on the ability of diagnosing skin lesions, both clinically and dermoscopically, is certainly of major importance for the dermatology clinic, a study like this is only of use if (1) the right methodology is applied and (2) the data are interpreted without any bias. Thus, in accordance with the expert reviewers I have some conceptual and technical concerns that need to be addressed. Rather than repeat further points here, I refer you to the specific remarks (below) for details. 

Reviewers' comments:

Reviewer's Responses to Questions

**Comments to the Author**

1. Is the manuscript technically sound, and do the data support the conclusions?

Reviewer #1: Yes

Reviewer #2: Partly

2. Has the statistical analysis been performed appropriately and rigorously? 

Reviewer #1: I Don't Know

Reviewer #2: Yes

3. Have the authors made all data underlying the findings in their manuscript fully available?

Reviewer #1: Yes

Reviewer #2: No

4. Is the manuscript presented in an intelligible fashion and written in standard English?

Reviewer #1: Yes

Reviewer #2: No

5. Review Comments to the Author

Reviewer #1: Disclaimer: I am not a statistician and this manuscript should be reviewed by a statistician

Title: This should include “…by naked-eye examination” because in some countries dermatoscopic evaluation is standard of care for any assessment of pigmented skin lesions by any practitioner

Design

What is the duration of a “dermatology rotation” and what proportion of that rotation involves assessing skin lesions with respect to melanoma diagnosis?

Were the readers asked if they were colour-vision impaired and excluded for that reason and were any discovered on Ishihara testing? I ask this because in a cohort of 152 it might be expected that some males at least would have an impairment. I know that does vary with ethnicity but it is worth commenting on.

Discussion

I suggest including the following observation in the discussion re Figure 1:

In C the lesion is green and so is the background and in D the lesion is red and the background pink, whereas in B (deuteranope) the surrounding skin is similar to that of the unadjusted image (A)

As a result the colour contrast between the lesion and the surrounding skin is greater for Figure 1B than for C and D which makes irregular structures at the border (right side) more visible (similar as in A). This makes the lesion more clearly asymmetrical which may lean the reader towards a malignant diagnosis.

(The Figure legend should include the appropriate letter for each image respectively)

It would be interesting to see more images. At least a collage similar to Figure 1 but with a malignant lesion. This may or may not support what I have stated about the contrast between lesion and surrounding skin colour. I suggest that the authors evaluate this with every lesion and include those results in the manuscript.

Additional Limitations:

The readers were all inexperienced and an undefined proportion were not practising clinicians. There was a selection bias for suspicious lesions which may have influenced the readers (all lesions had been excised). This could in part explain the lower overall accuracy for benign lesions.

Reviewer #2: Dear Authors,

thank you for your manuscript "Assessing the impact of color blindness on the ability of identifying benign and malignant skin lesions". The idea behind your study is appreciated. However, there are some important issues to be adressed.

1) please improve the writing, there are many errors to be corrected (example: Page 1. Background: Color vision deficiencydescribes the inability to distinguish certain

shades of color. We found a significantlyhigher mean accuracyasthe accuracy of classifying)

2) diagnostic precision. You cite several sources to emphasize that "The diagnosis and referral accuracy of pigmented skin lesions are generally low among primary care physicians compared to dermatologists [12–14]." Despite that fact your study was conducted with participant in their final year in medical school or newly graduated. Please describe your intention to choose this group of participants and discuss a potential impact on the results.

3) "we used Vischeck color blindness simulator in Fiji software [16], which has been proved to be highly accurate in simulating color-blind images [17]".

It remains unclear how the whole process was conducted. Please specify how the images were presented (probably online via a private account shown on an uncertified screen). Especially the presentation of colors may vary and is dependent from many factors (model and age of the screen, contrast, intensity, surroundings etc). This might be crucial regarding the results.

Best regards and good luck!

6. PLOS authors have the option to publish the peer review history of their article (what does this mean?). If published, this will include your full peer review and any attached files.

Reviewer #1: No

Reviewer #2: No

---

## [Author Response · Author response to Decision Letter 0]

11 Sep 2021

Reviewers' comments:

Reviewer's Responses to Questions

Comments to the Author

1. Is the manuscript technically sound, and do the data support the conclusions?

Reviewer #1: Yes

Reviewer #2: Partly

Response: Thank you.

2. Has the statistical analysis been performed appropriately and rigorously? 

Reviewer #1: I Don't Know

Reviewer #2: Yes

Response: The statistical analysis was performed by an expert biostatistician.

3. Have the authors made all data underlying the findings in their manuscript fully available?

Reviewer #1: Yes

Reviewer #2: No

Response: The data used in this study is openly and freely accessible at http://www.cs.rug.nl/~imaging/databases/melanoma_naevi/ with no restriction.

4. Is the manuscript presented in an intelligible fashion and written in standard English?

Reviewer #1: Yes

Reviewer #2: No

Response: We performed further English language polishing to ensure the absence of any typographical or grammatical errors.

5. Review Comments to the Author

Reviewer #1: Disclaimer: I am not a statistician and this manuscript should be reviewed by a statistician

Title: This should include “…by naked-eye examination” because in some countries dermatoscopic evaluation is standard of care for any assessment of pigmented skin lesions by any practitioner

Response: We agree with the reviewer, and we amended the title and further detailed this in the aim accordingly.

Design

What is the duration of a “dermatology rotation” and what proportion of that rotation involves assessing skin lesions with respect to melanoma diagnosis?

Response: We provided details on dermatology rotations at Jordanian universities: “all Jordanian universities usually have a 2-week long dermatology rotation with an exam at the end of rotation to assess acquired knowledge.”. Moreover, we provided a link for an optional educational slides on differentiating benign and malignant skin lesions:

https://reference.medscape.com/features/slideshow/suspicious-skin-lesions

Were the readers asked if they were colour-vision impaired and excluded for that reason and were any discovered on Ishihara testing? I ask this because in a cohort of 152 it might be expected that some males at least would have an impairment. I know that does vary with ethnicity but it is worth commenting on.

Response: We agree with the reviewer, we further explained this in the methodology, where any participant who had an Ishihara score below 12/13 should not proceed with the questionnaire.

Discussion

I suggest including the following observation in the discussion re Figure 1:

In C the lesion is green and so is the background and in D the lesion is red and the background pink, whereas in B (deuteranope) the surrounding skin is similar to that of the unadjusted image (A) As a result the colour contrast between the lesion and the surrounding skin is greater for Figure 1B than for C and D which makes irregular structures at the border (right side) more visible (similar as in A). This makes the lesion more clearly asymmetrical which may lean the reader towards a malignant diagnosis.

Response: We would like to than the reviewer for this important observation, which led us to more literature digging to find the basis to explain the observation. We found that the distinction of borders can be affected by colors at the border. However, the distinction is higher for high wavelength colors (i.e., red and green), and the distinction is lower for low-wavelength colors (i.e., blue). We provided this to the discussion to further explain this important point:

“It was noted that the borders between the lesion and the surrounding skin is less distinct for deuteranopia-simulated images, which might increase rater’s suspicion toward malignant lesion. This effect was also observed, but with a lesser degree, for protanopia-simulated images, but not for tritanopia-simulated images. This can be explained by previous studies’ observations, where the distinction of image borders increases for higher wavelength colors (i.e., toward the red colors) [19,20], and deuteranopia- and protanopia-simulated images have more low-wavelength colors than normal or tritanopia-simulated images.”.

(The Figure legend should include the appropriate letter for each image respectively)

Response: We added the letters referring to each simulation type.

It would be interesting to see more images. At least a collage similar to Figure 1 but with a malignant lesion. This may or may not support what I have stated about the contrast between lesion and surrounding skin colour. I suggest that the authors evaluate this with every lesion and include those results in the manuscript.

Response: We provided detailed discussion on the impact of contrast. As per journal’s request, we had to replace the image used from the database with a one from our own patients in order to publish the image under CC BY 4.0 license, so it was difficult to include an image for a malignant lesion. 

Additional Limitations:

The readers were all inexperienced and an undefined proportion were not practising clinicians. There was a selection bias for suspicious lesions which may have influenced the readers (all lesions had been excised). This could in part explain the lower overall accuracy for benign lesions.

Response: We agree with the reviewer, and we explained this limitation at the end of discussion as follows:

“Moreover, this study was not performed on specialized dermatologist, whom would have more experience and confidence in diagnosing benign lesions as benign depending on more lesion characteristic, other than color. Further studies in this regard might show if including experienced dermatologist might abolish such accuracy difference for participants with color vision deficiency.”

Reviewer #2: Dear Authors,

thank you for your manuscript "Assessing the impact of color blindness on the ability of identifying benign and malignant skin lesions". The idea behind your study is appreciated. However, there are some important issues to be adressed.

1) please improve the writing, there are many errors to be corrected (example: Page 1. Background: Color vision deficiency describes the inability to distinguish certain

shades of color. We found a significantlyhigher mean accuracyasthe accuracy of classifying)

Response: We reviewed the manuscript for typographical or grammatical errors. The above-mentioned typographical error appeared due to pdf-HTML conversion by the editorial system.

2) diagnostic precision. You cite several sources to emphasize that "The diagnosis and referral accuracy of pigmented skin lesions are generally low among primary care physicians compared to dermatologists [12–14]." Despite that fact your study was conducted with participant in their final year in medical school or newly graduated. Please describe your intention to choose this group of participants and discuss a potential impact on the results.

Response: This is an important point to elaborate. We explained the rationale of including participant in their final year in medical school or newly graduated in the introduction as follows: “We conducted this study on participant in their final year in medical school or newly graduated, whom we believe depends more on lesion color criteria in distinguishing benign and malignant lesions. Moreover, it also enabled us to include a relatively larger sample size.”. However, including such sample resulted in a limitation compared to including experienced dermatologist, which we also detailed in study limitations as follows: “Moreover, this study was not performed on specialized dermatologist, whom would have more experience and confidence in diagnosing benign lesions as benign depending on more lesion characteristic, other than color. Further studies in this regard might show if including experienced dermatologist might abolish such accuracy difference for participants with color vision deficiency.”.

3) "we used Vischeck color blindness simulator in Fiji software [16], which has been proved to be highly accurate in simulating color-blind images [17]".

It remains unclear how the whole process was conducted. Please specify how the images were presented (probably online via a private account shown on an uncertified screen). Especially the presentation of colors may vary and is dependent from many factors (model and age of the screen, contrast, intensity, surroundings etc). This might be crucial regarding the results.

Response: We agree with the reviewer on the importance of elaboration on this part. We used Google forms for data collection. To account for device variability, we assessed data about skin lesion classification without simulation, in order to compare with classification accuracy for simulated images. However, as this does not eliminate the impact of device screen variability, we also added a limitation statement in this regard: “Another limitation to keep in mind is that participants used their own devices to complete the questionnaire, which might result in impact of screen color characteristic and brightness on classification accuracy.”

Best regards and good luck!

6. PLOS authors have the option to publish the peer review history of their article (what does this mean?). If published, this will include your full peer review and any attached files.

Do you want your identity to be public for this peer review? For information about this choice, including consent withdrawal, please see our Privacy Policy.

Reviewer #1: No

Reviewer #2: No

---

## [Decision Letter · Decision Letter 1]

10 Jan 2022

PONE-D-21-20492R1

Assessing the Impact of Color Blindness on the Ability of Identifying Benign and Malignant Skin Lesions by Naked-Eye Examination

PLOS ONE

Dear Dr. AlRyalat,

Thank you for submitting your manuscript to PLOS ONE. After careful consideration, we feel that it has merit but does not fully meet PLOS ONE’s publication criteria as it currently stands. Therefore, we invite you to submit a revised version of the manuscript that addresses the points raised during the review process.

In agreement with the expert reviewers, the authors have improved the manuscript in the revised version. However, there are still points that need to be addressed. Please see the reviewers' comments below for detail.

We look forward to receiving your revised manuscript.

Kind regards,

Nikolas K. Haass, MD/PhD

Academic Editor

PLOS ONE

Reviewers' comments:

Reviewer's Responses to Questions

**Comments to the Author**

1. If the authors have adequately addressed your comments raised in a previous round of review and you feel that this manuscript is now acceptable for publication, you may indicate that here to bypass the “Comments to the Author” section, enter your conflict of interest statement in the “Confidential to Editor” section, and submit your "Accept" recommendation.

Reviewer #1: (No Response)

Reviewer #2: All comments have been addressed

2. Is the manuscript technically sound, and do the data support the conclusions?

Reviewer #1: Partly

Reviewer #2: Yes

3. Has the statistical analysis been performed appropriately and rigorously? 

Reviewer #1: I Don't Know

Reviewer #2: Yes

4. Have the authors made all data underlying the findings in their manuscript fully available?

Reviewer #1: Yes

Reviewer #2: Yes

5. Is the manuscript presented in an intelligible fashion and written in standard English?

Reviewer #1: No

Reviewer #2: Yes

6. Review Comments to the Author

Reviewer #1: See attachment for necessary formatting (bold italics re English grammar issues)

See attachment for necessary formatting (bold italics re English grammar issues)

Reviewer #2: Dear Authors,

thank you for improving your manuscript significantly. "We conducted this study on participant in their final year in medical school or newly graduated, whom we believe depends more on lesion color criteria in distinguishing benign and malignant lesions. Moreover, it also enabled us to include a relatively larger sample size." As far as I know the first statement is a hypothesis. The motivation to include a larger sample size might not justify to conduct such a study with graduating medical students instead of general practicioners or dermatolists. From a clinical stanpoint, conducting this study with general pracitioners would probably be ideal in order to simulate what you wanted to express.

7. PLOS authors have the option to publish the peer review history of their article (what does this mean?). If published, this will include your full peer review and any attached files.

Reviewer #1: No

Reviewer #2: **Yes: **Alexander F. Scheuerle, MD

---

## [Author Response · Author response to Decision Letter 1]

22 Jan 2022

Reviewer #1: 

Page 16: The following confusing statement must be clarified:

“We calculated the score out of 5 for each of benign and malignant lesion images

correctly identified in each of the four sets and calculated the accuracy score as a percentage for each.” At face value this states that each image is scored out of 5 so there would be a maximum score of 50 for each set of 10 images.

The next sentence, on face value, describes a second calculation.

This does not make sense to me.

Reply: We agree with the reviewer that the statement was not clear, so we rewrote the statement as follows:

“Each image’s answer was scored as either correct (1) or incorrect (0). There were four sets of images, each containing images for five benign and five malignant lesions. Of the four sets, 3 were simulated into into what a deuteranope, protanope, and tritanope participan can see, and the last set was not simulated. We calculated accuracy score as a percentage for the benign and malignant images in each set, then accuracy of image classification in the set (aka., overall accuracy).”

Page 18: The following sentence does not make sense:

Table 2 details the mean accuracy of diagnosing benign and malignant lesions for each group the difference in accuracy between benign and malignant lesion classification. (the same sentence is repeated in the legend for Table 2.

Response: We rewrote the statement and figure legend to be better descriptive for the results: 

“the mean accuracy of classifying lesions presented in the images in each set (i.e., protanope, deuteranope, tritanope simulated images and non-simulated images) as either benign or malignant lesions. It also present the mean difference in classification accuracy between images containing benign and malignant lesions.”

Table 2 presents the same data as Table 1 for benign and malignant accuracy, including values and Standard Deviation and essentially just adds a decimal point to the values and adds p-value, Mean Difference and 95% CI. This could be presented in a single table. An inconsistency between the two tables for malignant lesion accuracy for Deuteranope images is noted (80.6% vs. 82.58%)

Response: Thank you for the suggestion. We combined both tables into one as suggested and rechecked the numbers presented for accuracy.

The legend of Table 3 is not a complete sentence. I actually see no purpose in including Table 3 at all. The preceding sentence conveys the negative findings adequately.

Response: We completed the figure legend. Despite the absence of significant results in the table, we think it provides important descriptive data to be considered in gender comparison. 

Discussion

Page 20: There is a major typo: This study tested the accuracy of diagnosing benign and malignant pigmented skin lesions among simulated images as if seen by patients with different types…

Response: Corrected, thank you.

Re this statement: “Upon performing our sub-group analysis to analyze if there was a difference in diagnosis accuracy for benign or malignant lesions, we found that the diagnostic accuracy for benign lesions were significantly lower for all images, with highest magnitude for deuteranopia simulated images” it is actually noted that non-simulated images were associated with a lower accuracy for benign lesions than both simulated groups except Deuteranope. This must be acknowledged and discussed.

Response: Thank you for this observation. We acknowledged this finding, however, it might be difficult to further elaborate on it due to the small difference in benign lesion diagnosis accuracy between non-simulated, and those protanopia and tritanopia simulated images.

Conclusion

The study was not on distinguishing skin lesions, as stated , but on distinguishing images of skin lesions. That is significant and must be stated in the conclusion.

Response: We edited the conclusion and the manuscript throughout to ensure correct wording used.

The images presented in the revised manuscript (Figure 1) bear no reasonable resemblance at all to any skin lesion, benign or malignant.

Response: The database containing images used in the survey was provided in its respective database: http://www.cs.rug.nl/~imaging/databases/melanoma_naevi/

Due to copyright for the database, we can’t use any of its images as a figure. We obtained images for a benign (as in the figure 1), and for a malignant lesion (new figure 2) to show how simulation works. We agree with the reviewer that using images from the original database might be more reflective to what we used in this study, however, we could not obtain copyrights for database’s images. 

English Grammar Issues

Page 12: Classification accuracy for malignant lesions were higher than…

Page 14: We conducted this study on participants in their final year in medical

school or newly graduated, whom we believe depends more on lesion color …

The study included participants who were either at their final year in medical school or newly graduated and were working as general practitioners.

Page 15: Using an online form, the nature of the project described and a consent to participate signed

Page 16: In order to correct for device’s screen variabilities, we compared the resulted

accuracy for none simulated images with simulated images.

(Fig 1) shows a benign skin lesion simulated into what a protanope, deuteranope, and tritanope participants can see. The 40 images were shuffled and presented sequentially each image alone.

Page 18: Upon comparing classification accuracy for benign and malignant lesions between each image simulation, we found significant differences for all images, where classification accuracy for malignant lesions were higher than classification accuracy for benign lesions

Page 20: Upon performing our sub-group analysis to analyze if there was a difference in

diagnosis accuracy for benign or malignant lesions, we found that the diagnostic accuracy for

benign lesions were significantly lower for all images, with highest magnitude for deuteranopia simulated images

Page 21: Many efforts has been evaluated…

Page 22: …it still has limitations that need to be considered upon result’s interpretation

Page 22: clinical decision to decide to wither a lesion is benign or…

Page 22: Moreover, this study was not performed on specialized dermatologist, whom would have more experience and confidence in diagnosing benign lesions as benign depending on more lesion characteristic, other than color

Page 22: Further studies in this regard might show if including experienced dermatologist might

Response: Thank you for going through the manuscript for English language. We went through the manuscript to correct the pointed mistakes and other missed ones.

Reviewer #2:

Thank you for improving your manuscript significantly. "We conducted this study on participant in their final year in medical school or newly graduated, whom we believe depends more on lesion color criteria in distinguishing benign and malignant lesions. Moreover, it also enabled us to include a relatively larger sample size." As far as I know the first statement is a hypothesis. The motivation to include a larger sample size might not justify to conduct such a study with graduating medical students instead of general practicioners or dermatolists. From a clinical stanpoint, conducting this study with general pracitioners would probably be ideal in order to simulate what you wanted to express.

Response: Thank you for this important point. We edited the manuscript accordingly.

---

## [Editor Report · Decision Letter 2]

27 Jan 2022

PONE-D-21-20492R2

Assessing the Impact of Color Blindness on the Ability of Identifying Benign and Malignant Skin Lesions by Naked-Eye Examination

PLOS ONE

Dear Dr. AlRyalat,

Thank you for submitting your manuscript to PLOS ONE. After careful consideration, we feel that it has merit but does not fully meet PLOS ONE’s publication criteria as it currently stands. Therefore, we invite you to submit a revised version of the manuscript that addresses the points raised during the review process.I'd like to thank the authors for improving the manuscript. However, there are still some points that need to be addressed: (1)
The point of the study is to assess the impact of colour deficiency on diagnosis of photos of pigmented skin lesions. Therefore, Figures 1 and 2 should be representative of the examined dataset and should thus be replaced with better examples (clinically). Secondly, the quality should be similar or better than that of the Figure in the original submission (the image quality of the new images is not sufficient), unless the quality of the actual dataset was similarly low as that of the new figures. The figure legend should contain the given diagnosis (e.g. compound naevus and superficial spreading melanoma). Moreover, and importantly, it should be made clear in text and legend that the lesions in Figures 1 and 2 are NOT included in the set of assessed lesions.  (2)
‘Each image’s answer’ should be corrected to ‘The diagnosis for each image’ (3)
‘aka. overall accuracy’ should be corrected to ‘i.e. overall accuracy’ (4)
Table 2 should state ‘between males and female participants’ rather than ‘between males and females’ (5)
‘rater’s suspicion toward malignant’ should be ‘rater’s bias toward malignant’ (6)
There are still multiple semantic and/or grammatical errors.

We look forward to receiving your revised manuscript.

Kind regards,

Nikolas K. Haass, MD/PhD

Academic Editor

PLOS ONE

Journal Requirements:

Additional Editor Comments::

I'd like to thank the authors for improving the manuscript. However, there are still some points that need to be addressed:

(1) The point of the study is to assess the impact of colour deficiency on diagnosis of photos of pigmented skin lesions. Therefore, Figures 1 and 2 should be representative of the examined dataset and should thus be replaced with better examples (clinically). Secondly, the quality should be similar or better than that of the Figure in the original submission (the image quality of the new images is not sufficient), unless the quality of the actual dataset was similarly low as that of the new figures. The figure legend should contain the given diagnosis (e.g. compound naevus and superficial spreading melanoma). Moreover, and importantly, it should be made clear in text and legend that the lesions in Figures 1 and 2 are NOT included in the set of assessed lesions.

(2) ‘Each image’s answer’ should be corrected to ‘The diagnosis for each image’

(3) ‘aka. overall accuracy’ should be corrected to ‘i.e. overall accuracy’

(4) Table 2 should state ‘between males and female participants’ rather than ‘between males and females’

(5) ‘rater’s suspicion toward malignant’ should be ‘rater’s bias toward malignant’

(6) There are still multiple semantic and/or grammatical errors.
---

## [Author Response · Author response to Decision Letter 2]

15 Feb 2022

Response letter

Dear editor,

We are glad to submit our revised manuscript entitled:

Assessing the Impact of Color Blindness on the Ability of Identifying Benign and Malignant Skin Lesions by Naked-Eye Examination

Thank you for submitting your manuscript to PLOS ONE. After careful consideration, we feel that it has merit but does not fully meet PLOS ONE’s publication criteria as it currently stands. Therefore, we invite you to submit a revised version of the manuscript that addresses the points raised during the review process.

I'd like to thank the authors for improving the manuscript. However, there are still some points that need to be addressed:

(1) The point of the study is to assess the impact of colour deficiency on diagnosis of photos of pigmented skin lesions. Therefore, Figures 1 and 2 should be representative of the examined dataset and should thus be replaced with better examples (clinically). Secondly, the quality should be similar or better than that of the Figure in the original submission (the image quality of the new images is not sufficient), unless the quality of the actual dataset was similarly low as that of the new figures. The figure legend should contain the given diagnosis (e.g. compound naevus and superficial spreading melanoma). Moreover, and importantly, it should be made clear in text and legend that the lesions in Figures 1 and 2 are NOT included in the set of assessed lesions. 

Reply: Thank you for the clearly explained comment. We the main issue in the original image was the copyright of its deposition did not meet the copyright requirements of the Plos One journal, so we could not include one of the images used in the dataset as a figure in the manuscript. We made this point clear as requested in the figure legend. We discarded the previously included Fig.1 due to its low quality and we replaced it with a better figure with an improved quality and representation of the original dataset. The malignant lesion has similar characteristics to the images present within the MED_NOD dataset, so we just worked to improve its quality to meet journal’s requirements.

(2) ‘Each image’s answer’ should be corrected to ‘The diagnosis for each image’

Response: Done, thank you.

(3) ‘aka. overall accuracy’ should be corrected to ‘i.e. overall accuracy’

Response: Done, thank you.

(4) Table 2 should state ‘between males and female participants’ rather than ‘between males and females’

Response: Done, thank you.

(5) ‘rater’s suspicion toward malignant’ should be ‘rater’s bias toward malignant’

Response: Done, thank you.

(6) There are still multiple semantic and/or grammatical errors.

Response: We sent the manuscript for a professional medical English language professional.

---

## [Editor Report · Decision Letter 3]

7 Jun 2022

PONE-D-21-20492R3Assessing the Impact of Color Blindness on the Ability of Identifying Benign and Malignant Skin Lesions by Naked-Eye ExaminationPLOS ONE

Dear Dr. AlRyalat,

Thank you for submitting your manuscript to PLOS ONE. After careful consideration, we feel that it has merit but does not fully meet PLOS ONE’s publication criteria as it currently stands. Therefore, we invite you to submit a revised version of the manuscript that addresses the points raised during the review process.

Thank you very much for sending us the original sets of images used in the study. These photos certainly made a difference to the reviewers and me. In fact, I had two lengthy phone conversations with the reviewers today. As you can imagine from our previous communication, it would still be best to publish all 40 images of the test set as supplementary data, as this would allow not only us but also the reader to assess them and therefore make the paper more valuable to the community. Thus, I would encourage you (possibly together with the PLOS ONE editorial team) to have another strong attempt to get permission from the source of the photographs to publish them as supplementary information. If this is not possible, you should make a clear statement in the paper that the photographs are available from you on request at any time (the same way as you provided them to us privately). In any case, you will need to change the figures in the manuscript, as the point of the study is to assess the impact of colour deficiency on diagnosis of photos of pigmented skin lesions. Therefore, Figures 1 and 2 should be similar to the examined dataset and should thus be replaced with better examples. Figure 1 (benign lesion) looks reasonably similar to the benign lesions in the test sets. However, Figure 2 (melanoma) is structurally so different from Figure 1 that even a black-and-white image would suffice to see the difference (i.e. the difference between Figure 1 and 2 in the current version of the manuscript PONE-D-21-20492R3 is not relevant for the message of the study). Hence, a melanoma that is more similar to a naevus or a naevus that is more similar to a melanoma should be picked for illustration in Figures 1 and 2, to reflect the more subtle colour differences (relevant for colour deficient assessors) between benign and malignant lesions, rather than the less colour-dependent structural differences. Moreover, and importantly, it should be made clear in text and legend that the lesions in Figures 1 and 2 are NOT included in the set of assessed lesions.

We look forward to receiving your revised manuscript.

Kind regards,

Nikolas K. Haass, MD/PhD

Academic Editor

PLOS ONE

Journal Requirements:

Additional Editor Comments (if provided):

Dear Dr. AlRyalat, please accept my apologies for this delay and thank you very much for sending us the original sets of images used in the study. These photos certainly made a difference to the reviewers and me. In fact, I had two lengthy phone conversations with the reviewers today. As you can imagine from our previous communication, it would still be best to publish all 40 images of the test set as supplementary data, as this would allow not only us but also the reader to assess them and therefore make the paper more valuable to the community. Thus, I would encourage you (possibly together with the PLOS ONE editorial team) to have another strong attempt to get permission from the source of the photographs to publish them as supplementary information. If this is not possible, you should make a clear statement in the paper that the photographs are available from you on request at any time (the same way as you provided them to us privately). In any case, you will need to change the figures in the manuscript, as the point of the study is to assess the impact of colour deficiency on diagnosis of photos of pigmented skin lesions. Therefore, Figures 1 and 2 should be similar to the examined dataset and should thus be replaced with better examples. Figure 1 (benign lesion) looks reasonably similar to the benign lesions in the test sets. However, Figure 2 (melanoma) is structurally so different from Figure 1 that even a black-and-white image would suffice to see the difference (i.e. the difference between Figure 1 and 2 in the current version of the manuscript PONE-D-21-20492R3 is not relevant for the message of the study). Hence, a melanoma that is more similar to a naevus or a naevus that is more similar to a melanoma should be picked for illustration in Figures 1 and 2, to reflect the more subtle colour differences (relevant for colour deficient assessors) between benign and malignant lesions, rather than the less colour-dependent structural differences. Moreover, and importantly, it should be made clear in text and legend that the lesions in Figures 1 and 2 are NOT included in the set of assessed lesions.

Reviewers' comments:

Included in Editor Comments.

---

## [Author Response · Author response to Decision Letter 3]

9 Jun 2022

We highly appreciate the time and effort Prof. Haass invested in our manuscript. 

First, we deposited the images used in this study, including simulated images, in a separate database and we provided citation and link to the database:

AlRyalat, Saif Aldeen, 2022, "Cutaneous benign and malignant lesions simulated for color blindness", https://doi.org/10.7910/DVN/OX324U, Harvard Dataverse, V1

Second, we replaced figure two, where we used a new image for a melanoma previously reported in a Plos One article:

“Longo C, Piana S, Lallas A, Moscarella E, Lombardi M, Raucci M, Pellacani G, Argenziano G. Routine clinical-pathologic correlation of pigmented skin tumors can influence patient management. PLoS One. 2015 Sep 1;10(9):e0136031”

This image has a CC-BY copyright license. We used this image and we provided proper citation accordingly.

Third, we reiterated on the fact that the figures presented in this article (figures 1 and 2) were not used as part of the survey used for data collection.

We hope these changes satisfy the merit for publication in Plos One Journal.

---

## [Editor Report · Decision Letter 4]

13 Jun 2022

Assessing the Impact of Color Blindness on the Ability of Identifying Benign and Malignant Skin Lesions by Naked-Eye Examination

PONE-D-21-20492R4

Dear Dr. AlRyalat,

We’re pleased to inform you that your manuscript has been judged scientifically suitable for publication and will be formally accepted for publication once it meets all outstanding technical requirements.

Kind regards,

Nikolas K. Haass, MD/PhD

Academic Editor

PLOS ONE

Additional Editor Comments (optional):

The authors have addressed all concerns.
---

## [Editor Report · Acceptance letter]

15 Jun 2022

PONE-D-21-20492R4 

Assessing the Impact of Color Blindness on the Ability of Identifying Benign and Malignant Skin Lesions by Naked-Eye Examination 

Dear Dr. AlRyalat:

I'm pleased to inform you that your manuscript has been deemed suitable for publication in PLOS ONE. Congratulations! Your manuscript is now with our production department. 

Kind regards, 

on behalf of

Prof Nikolas K. Haass 

Academic Editor

PLOS ONE